# The Impact of Severe Maternal Morbidity on Perinatal Outcomes in High Income Countries: Systematic Review and Meta-Analysis

**DOI:** 10.3390/jcm9072035

**Published:** 2020-06-29

**Authors:** Tesfaye S. Mengistu, Jessica M. Turner, Christopher Flatley, Jane Fox, Sailesh Kumar

**Affiliations:** 1Mater Research Institute, University of Queensland, Level 3 Aubigny Place, Raymond Terrace, South Brisbane, QLD 4101, Australia; t.mengistu@uq.net.au (T.S.M.); jessica.turner@uq.edu.au (J.M.T.); cjflatley@gmail.com (C.F.); 2School of Public Health, College of Medicine and Health Sciences, Bahir Dar University, Bahir Dar, P.O. Box 79, Ethiopia; 3Faculty of Medicine, The University of Queensland, Herston, QLD 4072, Australia; jane.fox@mater.uq.edu.au

**Keywords:** severe maternal morbidity, adverse perinatal outcomes, high income countries

## Abstract

While there is clear evidence that severe maternal morbidity (SMM) contributes significantly to poor maternal health outcomes, limited data exist on its impact on perinatal outcomes. We undertook a systematic review and meta-analysis to ascertain the association between SMM and adverse perinatal outcomes in high-income countries (HICs). We searched for full-text publications in PubMed, Embase, Cumulative Index of Nursing and Allied Health Literature (CINAHL), and Scopus databases. Studies that reported data on the association of SMM and adverse perinatal outcomes, either as a composite or individual outcome, were included. Two authors independently assessed study eligibility, extracted data, and performed quality assessment using the Newcastle–Ottawa Scale. We used random-effects modelling to calculate odds ratios (ORs) with 95% confidence intervals. We also assessed the risk of publication bias and statistical heterogeneity using funnel plots and Higgins I^2^, respectively. We defined sub-groups of SMM as hemorrhagic disorders, hypertensive disorders, cardiovascular disorders, hepatic disorders, renal disorders, and thromboembolic disorders. Adverse perinatal outcome was defined as preterm birth (before 37 weeks gestation), small for gestational age (SGA) (birth weight (BW) < 10th centile for gestation), low birthweight (LBW) (BW < 2.5 kg), Apgar score < 7 at 5 min, neonatal intensive care unit (NICU) admission, stillbirth and perinatal death (stillbirth and neonatal deaths up to 28 days). A total of 35 studies consisting of 38,909,426 women were included in the final analysis. SMMs associated with obstetric hemorrhage (OR 3.42, 95% CI: 2.55–4.58), severe hypertensive disorders (OR 6.79, 95% CI: 6.06–7.60), hepatic (OR 3.19, 95% CI: 2.46–4.13) and thromboembolic disorders (OR 2.40, 95% CI: 1.67–3.46) were significantly associated with preterm birth. SMMs from hypertensive disorders (OR 2.86, 95% CI: 2.51–3.25) or thromboembolic disorders (OR 1.48, 95% CI: 1.09–1.99) were associated with greater odds of having SGA infant. Women with severe hemorrhage had increased odds of LBW infant (OR 2.31, 95% CI: 1.57–3.40). SMMs from obstetric hemorrhage (OR 4.16, 95% CI: 2.54–6.81) or hypertensive disorders (OR 4.61, 95% CI: 1.17–18.20) were associated with an increased odds of low 5-min Apgar score and NICU admission (Severe obstetric hemorrhage: OR 3.34, 95% CI: 2.26–4.94 and hypertensive disorders: OR 3.63, 95% CI: 2.63–5.02, respectively). Overall, women with SMM were 4 times more likely to experience stillbirth (OR 3.98, 95% CI: 3.12–7.60) compared to those without SMM with cardiovascular disease (OR 15.2, 95% CI: 1.29–180.60) and thromboembolic disorders (OR 9.43, 95% CI: 4.38–20.29) conferring greatest risk of this complication. The odds of neonatal death were significantly higher in women with SMM (OR 3.98, 95% CI: 2.44–6.47), with those experiencing hemorrhagic (OR 7.33, 95% CI: 3.06–17.53) and hypertensive complications (OR 3.0, 95% CI: 1.78–5.07) at highest risk. Overall, SMM was also associated with higher odds of perinatal death (OR 4.74, 95% CI: 2.47–9.12) mainly driven by the increased risk in women experiencing severe obstetric hemorrhage (OR 6.18, 95% CI: 2.55–14.96). Our results highlight the importance of mitigating the impact of SMM not only to improve maternal health but also to ameliorate its consequences on perinatal outcomes.

## 1. Introduction

Maternal health is a key determinant of perinatal outcomes and an important indicator of a nation’s overall socioeconomic progress [1]. However, despite consistent and significant improvements in maternal health over the last century, there remains considerable global inequity in obstetric and neonatal health outcomes. Furthermore, although precipitous declines in maternal mortality rates have occurred in many countries, rates of severe maternal morbidity (SMM) have not shown similar improvements and are increasing in some regions, mainly because of high rates of pre-existing co-morbidities, including diabetes and obesity [2]. The causes of maternal morbidity are varied, complex and inter-related. The Maternal Morbidity Working Group led by the World Health Organization (WHO, Geneva, Switzerland) broadly defines it as “any health condition attributed to and/or aggravated by pregnancy and childbirth that has a negative impact on the woman’s wellbeing” [1] and has included this definition into the International Classification of Diseases—11 [3]. In contrast, SMM is usually defined as a “near miss” episode characterized as the near death of a woman surviving pregnancy, or a childbirth-related complication, or within 42 days of the termination of pregnancy [4,5]. The true global burden of SMM is not known and thus poorly understood, primarily because of the lack of a standardized definition as well as, crucially, inconsistent recording and reporting of this outcome [1,6,7,8].

Whilst maternal and perinatal mortality and morbidity are major health issues in low- and middle-income countries, even high-income countries (HICs) are not immune from this issue. In one North American study [9], SMM was approximately 50 times more frequent than maternal mortality, with massive post-partum hemorrhage and complications relating to hypertensive disease in pregnancy common antecedents of severe morbidity [9,10].

Surprisingly, the reduction in maternal mortality rates in HICs is associated with a paradoxical increase in the incidence of SMM which is driven by a combination of factors, including more advanced maternal age at first pregnancy, obesity, chronic medical co-morbidities, and rising rates of operative birth, particularly caesarean section [11]. SMM is now increasingly recognized as an important obstetric care quality indicator for which WHO recommends that HICs should have appropriate surveillance measures in order to identify trends and system failures. SMM is also an important risk factor for adverse perinatal outcomes [10,11,12], although precise estimates of the magnitude of its contributory risks are unclear, despite both sharing many similar causative pathways [13,14]. Against this background, the aim of this systematic review and meta-analysis was to detail the impact of SMM on perinatal outcomes in HICs.

## 2. Materials and Methods

This systematic review and meta-analysis was registered with the International Prospective Register of Systematic Reviews (PROSPERO) (ID: CRD42019130933) and conducted in accordance with a previously published protocol [15]. We searched PubMed, Embase, Cumulative Index of Nursing and Allied Health Literature (CINAHL), and Scopus using different combinations of key terms and search strategies with no restriction by year of publication (Appendix A). The reference lists of included studies were then checked to identify any additional relevant articles. We included only studies published in English, conducted in HICs and meeting the following criteria: (1) original studies of any design, (2) conducted on women with singleton pregnancies > 20 week’s gestation, (3) presented data/results on the association between SMM (as defined by the WHO maternal near-miss criteria) and adverse perinatal outcomes (either as a composite or by individual outcome) and reported sufficient information to calculate risk estimates. The World Bank’s definition of HICs was used [16]. We excluded studies of women with multiple pregnancy, pregnancies ending before 20 weeks of gestation, systematic reviews, case series/reports, conference papers/abstracts, proceedings, editorial reviews, letter of communications, commentaries, studies with a small sample size (*n* < 10) and qualitative studies.

Two reviewers independently assessed the eligibility of studies using the Population/participants, Interventions, Comparisons, Outcomes, and Study design (PICOS) framework [17]. The first reviewer (T.S.M) screened all citations by title, abstract and full text. A second reviewer (J.T) independently reviewed the titles, abstracts, and full text of the screened publications for eligibility. Disagreements on the screening and inclusion of studies were discussed and resolved by consensus with the assistance of a third reviewer (S.K). Findings are reported in accordance with the Preferred Reporting Items for Systematic Reviews and Meta-Analyses (PRISMA) guidelines [18].

The methodological quality of the included studies was assessed using the Newcastle–Ottawa scale (NOS) [19]. This tool consists of three domains: selection of study participants, comparability of study groups and reporting and determination of outcomes. Each study was graded out of nine points (separately for case-control and cohort studies) as per the NOS coding manual and summarized in three categories as good (if total score ≥ 7), fair (if total score 5–6) or poor (if total score < 5). Studies were deemed to be at high risk of bias if the NOS score was ≤ 6 [20]. 

Two authors (T.S.M. and J.T.) extracted study characteristics, the definition of SMM used in the study, details of adverse perinatal outcomes and other key study findings. We defined sub-groups of SMM as hemorrhagic disorders (postpartum hemorrhage, antepartum hemorrhage, bleeding of unknown origin, abnormally invasive placenta, uterine rupture and hysterectomy), hypertensive disorders (severe gestational hypertension, severe preeclampsia, eclampsia and hemolysis, elevated liver enzymes and low platelets (HELLP) syndrome), cardiovascular disorders (ischemic or hemorrhagic stroke, cardiac arrhythmia, peripartum cardiomyopathy and cardiac arrest), hepatic disorders, renal disorders and thromboembolic disorders (amniotic fluid embolism, pulmonary embolism, or deep venous thrombosis). We defined adverse perinatal outcome as: preterm birth (before 37 weeks gestation), small for gestational age (SGA) (birth weight (BW) < 10th centile for gestation), low birthweight (LBW) (BW < 2.5 kg), Apgar score < 7 at 5 min, neonatal intensive care unit (NICU) admission, stillbirth and perinatal death (stillbirth and neonatal deaths up to 28 days). 

### Statistical Analysis

We used Review Manager Software (RevMan; Version 5.3. Copenhagen: The Nordic Cochrane Centre, The Cochrane Collaboration, 2014) for data entry and statistical analysis. Heterogeneity between studies was assessed using Higgins I^2^ statistics and considered high if I^2^ ≥ 50% [21]. Because of the heterogeneity of maternal conditions contributing to SMM, as well as the variability within and between studies, we used the random-effects Mantel–Haenszel method to calculate study-specific and pooled odds ratios (ORs) with 95% confidence intervals. As far as possible, we categorized SMM according to the most likely underlying etiology—hemorrhagic, hypertensive, cardiovascular, hepatic, renal or thromboembolic disorders. The effect of publication bias was assessed using funnel plots. Sub-group analyses were performed according to the presumed etiology of SMM. Sensitivity analyses were performed by sequentially removing studies at high risk of bias to evaluate the impact of SMM on different adverse perinatal outcomes.

## 3. Results

### 3.1. Literature Search and Study Selection

The process of study screening, selection and reasons for exclusion are shown in Figure 1. Of the 18,434 studies identified by the initial search, 11,196 (after removing duplicates) were screened by title and 5181 were selected for further abstract screening. Of these, 35 original studies (containing a total of 38,909,426 participants) were eligible for full-text review and used for the final analysis (Table 1).

### 3.2. Characteristics of Included Studies and Risk of Bias

The characteristics of all included studies and their overall quality score are summarized in Table 2. Of the 35 studies, 26 [22,23,24,25,26,27,28,29,30,31,32,33,34,35,36,37,38,39,40,41,42,43,44,45,46,47] were population-based cohort studies, 8 were case-control studies [48,49,50,51,52,53,54,55] and 1 study [56] was a cross-sectional study. Fifteen studies were conducted in North America [22,27,28,29,30,33,34,35,39,45,47,50,54,55,56], four each in Australia [23,38,40,42] and Israel [24,41,43,44], three in Canada [31,36,37], two each in Finland [48,51], Sweden [32,46] and the United Kingdom [25,49] and one each in the Netherlands [26], South Korea [53] and Taiwan [52].

The methodological quality of the studies were rated as: good (21 studies) [23,24,25,26,27,28,34,35,36,37,38,39,40,44,48,49,50,51,52,55,56], fair (11 studies) [22,29,30,31,32,33,42,43,45,46,54] and poor (3 studies) [41,47,53]. Five studies had a NOS score < 6 [22,29,41,47,53] and were deemed to have high risk of bias. Most of the included studies scored high in the participant selection and outcome assessment categories, while most of the variability between studies was in the comparability category shown in Table 2. The pooled global effect, citations and Higgins I^2^ values for each sub-group of SMM and adverse perinatal outcomes are shown in Table 3. 

### 3.3. Meta-Analysis: Effect of SMM on Adverse Perinatal Outcomes

#### 3.3.1. Preterm Birth

This outcome was reported in 20 studies [22,23,24,25,26,27,29,31,32,34,37,38,40,42,46,47,49,51,52,54]. Overall, women who had SMM were three times more likely to experience preterm birth (OR 3.11; 95% CI: 2.56–3.78). However, the risk of preterm birth was variably influenced depending on the underlying etiology of SMM (Chi^2^ = 110.58, *p* < 0.0001, I^2^ = 95.50%). Obstetric hemorrhage (OR 3.42, 95% CI: 2.55–4.58), hypertensive (OR 6.79, 95% CI: 6.06–7.60), hepatic (OR 3.19, 95% CI: 2.46–4.13) and thromboembolic disorders (OR 2.40, 95% CI: 1.67–3.46) were all associated with increased odds of preterm birth. There was, however, no significant association between SMM and preterm birth for cardiovascular disorders (Figure 2).

#### 3.3.2. Small for Gestational Age Infant

SGA as an outcome was reported in 18 studies [22,23,24,26,27,29,30,31,37,40,44,46,47,48,49,50,52,54]. Although the pooled effect of SMM for SGA was not significant (OR 1.33, 95% CI: 0.98–1.81), women who had SMM associated with hypertensive disorders (OR 2.86, 95% CI: 2.51–3.25) or thromboembolic disorders (OR 1.48, 95% CI: 1.09–1.99) had greater odds of having a SGA infant. The effect of SMM on SGA also showed significant sub-group differences (Chi^2^ = 65.16, *p* < 0.0001, I^2^ = 92.30%) (Figure 3).

#### 3.3.3. Low Birth Weight

The association between SMM and LBW was assessed in 11 studies [23,31,34,38,41,43,44,48,51,52,55]. There was substantial heterogeneity in the results. The pooled effect demonstrated higher odds of LBW in women with SMM (OR 2.20, 95% CI: 1.56–3.09), with only severe hemorrhage exhibiting increased odds of this outcome (OR 2.31, 95% CI: 1.57–3.40) (Figure 4).

#### 3.3.4. Five-Minute Apgar Score < 7

Fifteen studies [23,24,28,29,38,41,43,44,45,46,48,49,51,53,54] reported this outcome. Pooled analysis showed higher odds of low 5-min Apgar score (OR 3.66, 95% CI: 2.41–5.56), albeit with considerable study variability (Chi^2^ = 228.26; I^2^ = 93%). Based on sub-group analysis, severe obstetric hemorrhage (OR 4.16, 95% CI: 2.54–6.81) and hypertensive disorders (OR 4.61, 95% CI: 1.17–18.20) were associated with increased odds of low 5-min Apgar score (Figure 5). 

#### 3.3.5. Admission to Neonatal Intensive Care Unit

The association between SMM and NICU admission was reported in 14 studies [23,25,26,27,28,29,34,37,38,45,49,51,53,55]. SMM was associated with increased odds of admission to NICU (OR 3.22, 95% CI: 2.45–4.25). There was, however, significant heterogeneity (Chi^2^ = 340.61; I^2^ = 97%) and sub-group differences (Chi^2^ = 30.30, I^2^ = 86.80%) within the eligible studies. Severe obstetric hemorrhage (OR 3.34, 95% CI: 2.26–4.94), hypertensive (OR 3.63, 95% CI: 2.63–5.02) and hepatic disorders (OR 1.89, 95% CI: 1.11–3.20) were associated with increased odds of NICU admission (Figure 6). 

#### 3.3.6. Stillbirth

Eighteen studies [22,23,25,27,29,31,33,34,37,38,40,42,46,49,51,53,54,56] reported the association between SMM and stillbirth. Women with SMM were about five times more likely to experience stillbirth (OR 4.87, 95% CI: 2.63–9.01) compared to those without SMM. Those with cardiovascular disease (OR 15.24, 95% CI: 1.29–180.60) or acute renal (OR 15.16, 95% CI: 4.41–52.12) or thromboembolic disorders (OR 5.07, 95% CI: 3.12–8.24) had significantly higher odds of this complication (Figure 7). 

#### 3.3.7. Neonatal Death

Thirteen studies [23,25,27,28,31,32,37,38,46,50,51,53,54] reported this outcome. The odds of neonatal death were significantly higher in women with SMM (OR 4.02, 95% CI: 2.45–6.59). There was, however, significant study variability (Chi^2^ = 125.28; I^2^ = 89%). We also found that the odds of this adverse outcome were greater in women with hemorrhagic (OR 7.33, 95% CI: 3.06–17.53) and hypertensive disorders (OR 3.00, 95% CI: 1.78–5.07). The overall test of sub-group difference was also significant with substantial heterogeneity (Chi^2^ = 15.33, *p* = 0.002, I^2^ = 80.4%) (Figure 8). 

#### 3.3.8. Perinatal Death

Ten studies [24,26,31,38,41,43,44,48,51,54] reported perinatal death. SMM was associated with higher odds of perinatal death (OR 4.74, 95% CI: 2.47–9.12). There was, however, significant sub-group difference (Chi^2^ = 10.16, *p* = 0.04, I^2^ = 60.60%), as well as substantial heterogeneity (Chi^2^ = 1.04; Chi^2^ = 98.75, *p* < 0.0001); I^2^ = 89%). Sub-group analysis confirmed that only obstetric hemorrhage was associated with greater odds of perinatal death (OR 6.18, 95% CI: 2.55–14.96) (Figure 9).

## 4. Discussion

The results of this systematic review and meta-analysis clearly show the strong and consistent association between SMM and adverse perinatal outcomes in women with singleton pregnancies in HICs. We found that women who experienced SMM were at significantly greater risk for preterm birth, SGA and LBW infants, low 5-min Apgar score, NICU admission, stillbirth, neonatal death, and overall perinatal death. Although we were unable to compare SMM by country due to the heterogeneity of definitions for SMM, our results are in concordance with other data showing that the association between SMM and adverse perinatal outcome does not appear to be influenced by differences in healthcare systems in these countries [11]. Our results highlight the crucial importance of mitigating SMM through high quality care given its impact not only on maternal health but also its consequences on perinatal outcomes. 

Our findings related to the compelling relationship between SMM and stillbirth (OR 4.87, 95% CI: 2.63–9.01) are consistent with another large study from North America [57] which showed that SMM was associated with a significantly higher odds of stillbirth after 23 weeks’ gestation (aOR 7.05, 95% CI: 6.27–7.93), particularly among women with other co-morbidities. There is also evidence that SMM is an independent risk factor for infant mortality (RR 1.39, 95% CI: 1.14–1.70) in very preterm infants, particularly in the first year of life [58]. 

Our broader results are consistent with other evidence [11] linking maternal morbidity and reproductive outcomes [59], thus highlighting the crucial importance of primary prevention and reducing the burden of SMM to mitigating the burden of SNM [2,60] and its associated consequences [10,12]. It is an imperative that transcends the socioeconomic status of a country. Whilst in many cases, SMM can rapidly develop and escalate, it is vital that healthcare professionals are aware of risk factors that predispose to poor outcomes, particularly in susceptible women. The perinatal risks we highlight are likely to be even more marked and impactful in low- and middle-income countries, and efforts should continue for effective surveillance of SMM rates as a measure of maternal health and perinatal outcome regardless of a nation’s socioeconomic development status. Our results highlight specific perinatal risks associated with a variety of maternal obstetric complications causing SMM and should be of benefit for clinicians. Indeed, there is evidence that almost 40% of cases of SMM are preventable [11]. 

To ascertain the impact of SMM, robust and ascertainable data are required. The WHO [11] and the American College of Obstetricians and Gynecologists [61] recommend careful surveillance using real-time data, which are sometimes lacking, even in HICs. However, despite these recommendations, there remains uncertainty as to the optimum surrogate indicators for SMM. Some HICs [62,63] have maternity outcome surveillance systems which use regular data updates to identify trends and adverse outcomes associated with SMM. However, although there is acknowledgement of rising rates of SMM [60] and the limitations of current surveillance systems [64], only a few HICs have instituted specific monitoring systems for SMM [62,63]. Developing a universally accepted classification system for SMM [61,65,66,67,68,69] would help in the standardized collection and reporting of important clinical data. In Europe, using hospital discharge data from eight countries, Chantry et al. found that diagnosis codes indicating obstetric hemorrhage, hysterectomy and red cell transfusion were all good candidates for the surveillance of maternal morbidity [70].

### Strengths and Limitations

One of the limitations of this review is the use of the WHO near miss criteria to define SMM, as these criteria are not consistently used by all HICs. Furthermore, we limited our analysis to only women with singleton pregnancies, cognizant that multiple pregnancy is an additional risk factor for adverse outcomes. Although the WHO near miss definition is widely accepted, not all HICs use this definition. The EURO-PERISTAT collaboration of 15 European countries defined SMM as a composite of the rates of eclampsia, hysterectomy for postpartum hemorrhage, ICU admission, blood transfusion, and uterine artery embolization [66], whilst the French EPIMOMS study group recommended 17 indicators (some which overlap with the EURO-PERISTAT indicators) specifically for use in HICs [65]. In the United States, a broadly similar list of 18 indicators is used [67,69]. 

In our systematic review, five studies were at high risk of publication bias, mainly because of a lack of confounder adjustment and comparison group ascertainment [22,29,41,47,53]. We also observed evidence of funnel plot asymmetry in some of the funnel plots. Interpretation of possible publication bias using funnel plots should take into account that funnel plots are crude and subjective [71], and are inaccurate measures [72] of publication bias, where its asymmetry does not necessarily indicate publication bias, and give misleading interpretations [73]. Our sensitivity analyses also demonstrate that the results were not significantly influenced by studies with high a risk of publication bias. We were unable to perform subgroup analysis based on study designs because of data limitations. Additionally, we chose to use odds ratios for our analyses, which may not always reflect the true risks at the population level. However, Viera et al., recommends that either odds ratios or relative risks are equally reliable when assessing rare events such as adverse perinatal outcomes [74].

Sample size variation between studies (64 women [50] to 12,524,119 women [56]) and study design, the use of heterogeneous SMM definitions, differences in participant characteristics and sampling procedures would also have introduced high heterogeneity into our analysis. However, our use of random-effects modeling [75] as well as the use of components within a widely accepted and standard SMM definition mitigates this limitation.

## 5. Conclusions

This systematic review and meta-analysis provide data demonstrating the robust association between SMM and adverse perinatal outcomes as well as highlighting specific maternal conditions that are risk factors for SMM and adverse perinatal outcomes. Our results also highlight an obvious research gap and emphasize the need for ongoing surveillance in all countries, regardless of socio-economic development status.

## Figures and Tables

**Figure 1 jcm-09-02035-f001:**
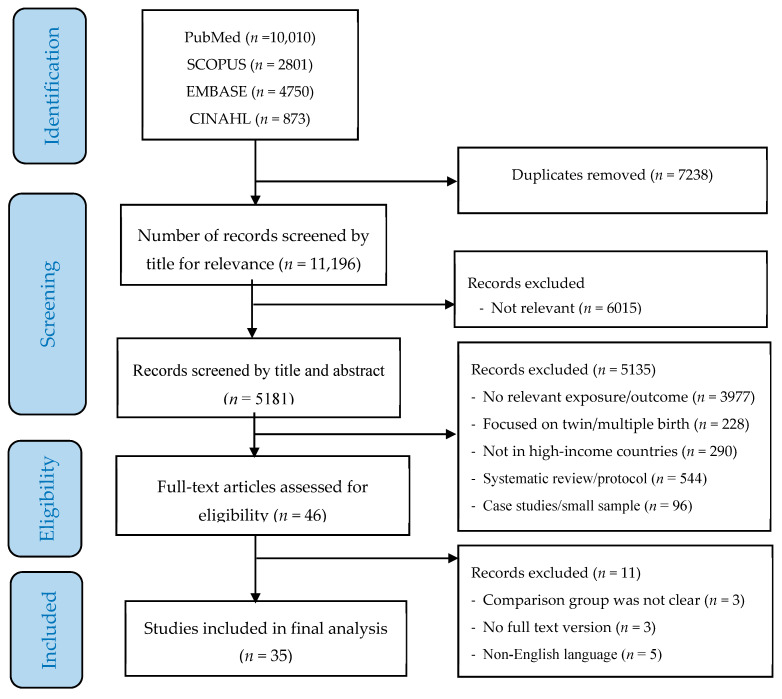
PRISMA flow diagram of screening and selection process.

**Figure 2 jcm-09-02035-f002:**
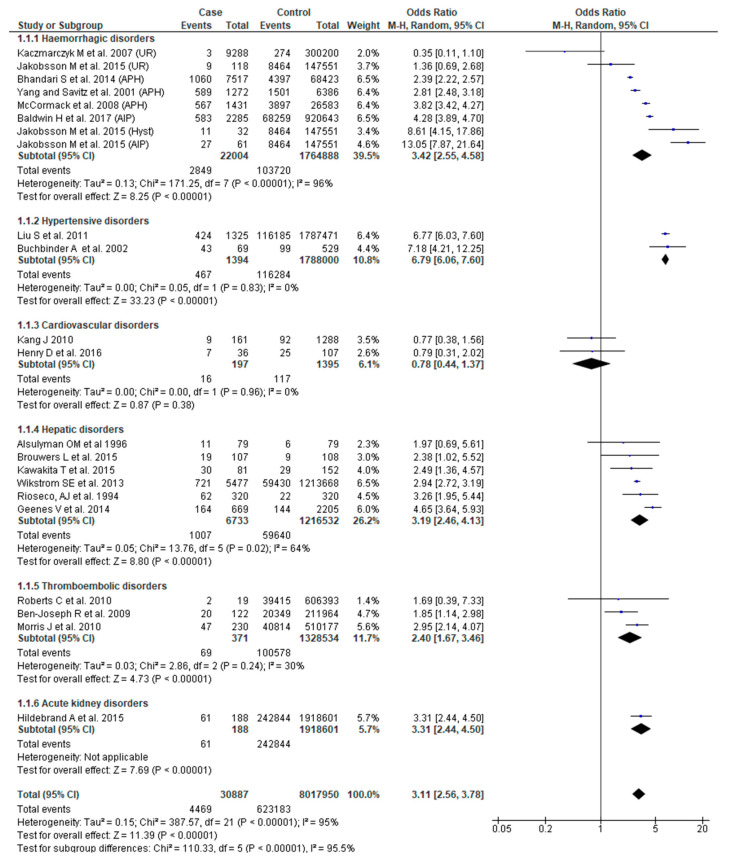
Forest plot of studies assessing association of Severe maternal morbidity (SMM) and preterm birth.

**Figure 3 jcm-09-02035-f003:**
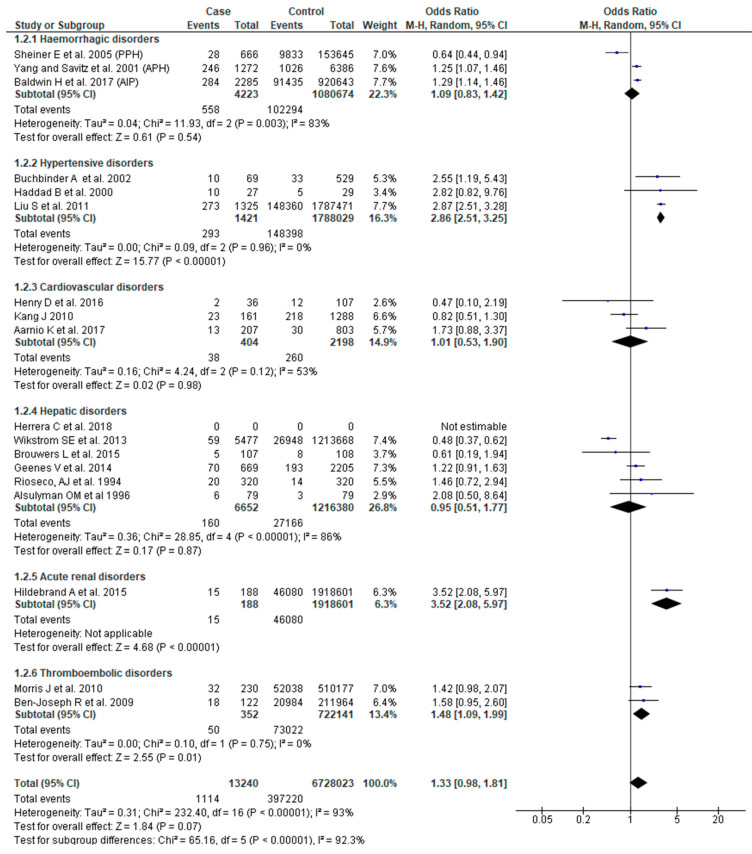
Forest plot of studies assessing association between SMM and infants being small for gestational age.

**Figure 4 jcm-09-02035-f004:**
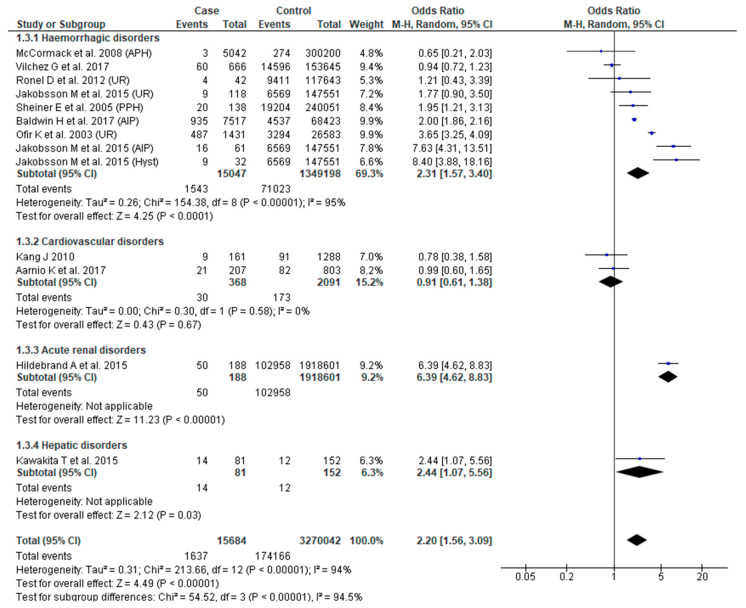
Forest plot of studies assessing association between Severe maternal morbidity (SMM) and low birth weight.

**Figure 5 jcm-09-02035-f005:**
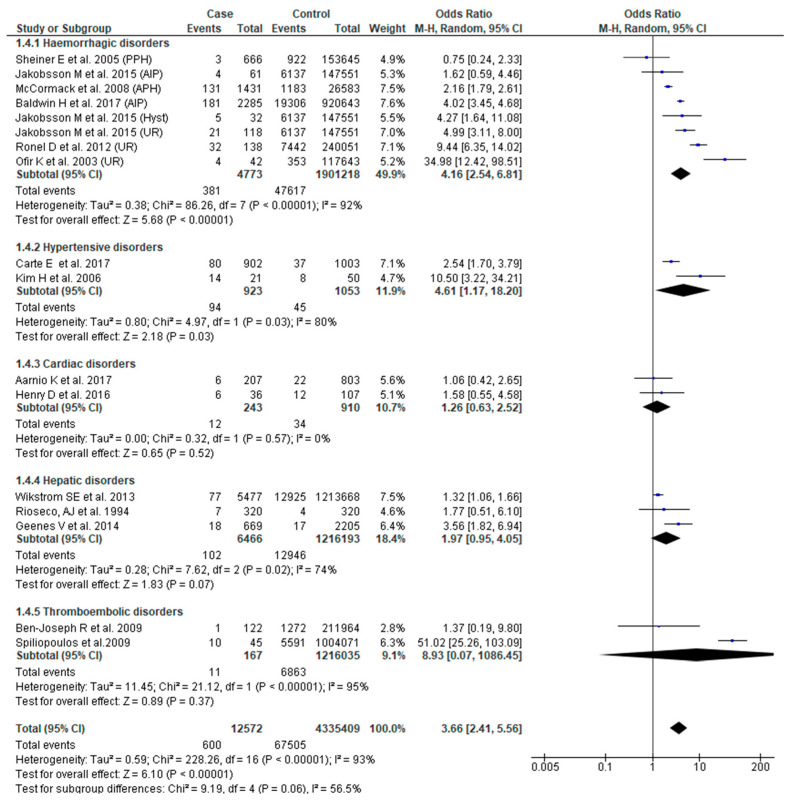
Forest plot of studies assessing association between SMM and 5-min Apgar score < 7.

**Figure 6 jcm-09-02035-f006:**
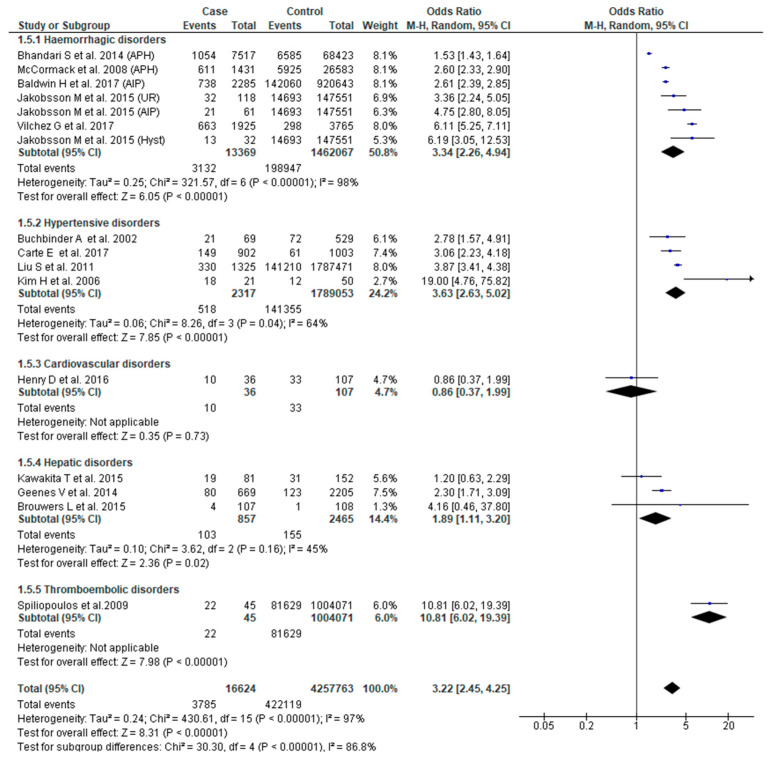
Forest plot of studies assessing association between Severe maternal morbidity (SMM) and neonatal intensive care unit (NICU) admission.

**Figure 7 jcm-09-02035-f007:**
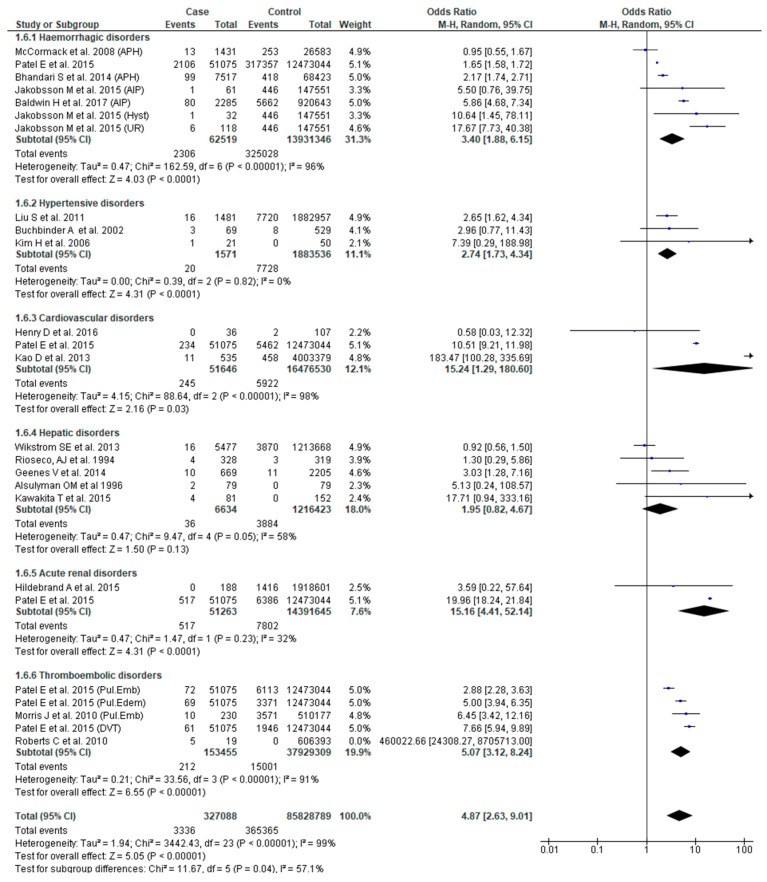
Forest plot of studies assessing association between Severe maternal morbidity (SMM) and stillbirth.

**Figure 8 jcm-09-02035-f008:**
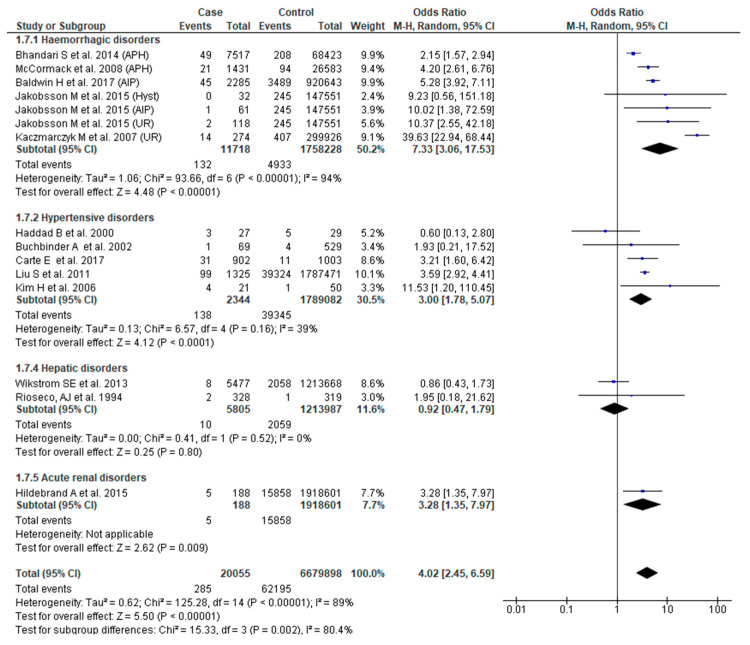
Forest plot of studies assessing association between Severe maternal morbidity (SMM) and neonatal death.

**Figure 9 jcm-09-02035-f009:**
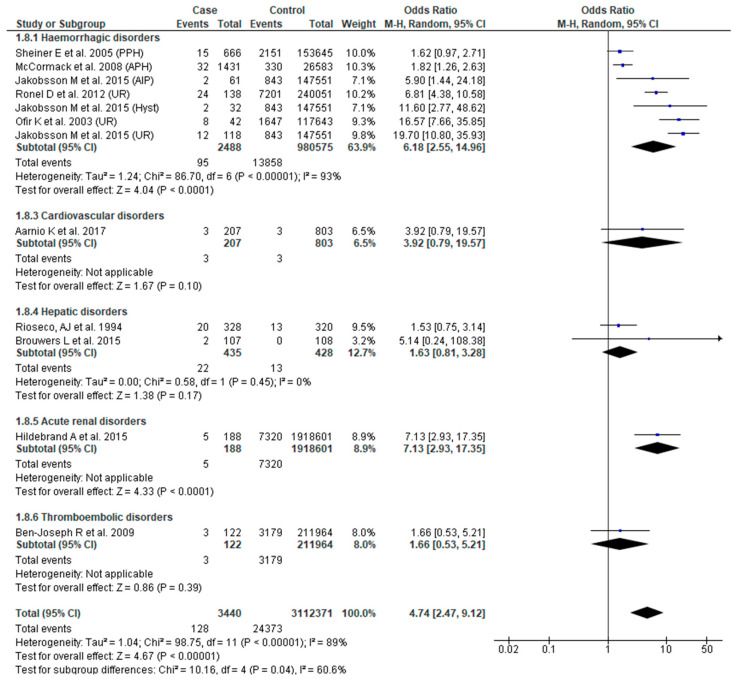
Forest plot of studies assessing association between Severe maternal morbidity (SMM) and perinatal death.

**Table 1 jcm-09-02035-t001:** Publications included in the systematic review.

Etiologic Subgroup of SMM	Country	Definition of SMM
**Severe hypertensive disorders**		
Buchbinder A. et al., 2002	USA	Severe gestational hypertension
		Severe preeclampsia
Carte E. et al., 2017	USA	Severe preeclampsia
Kim H. et al., 2006	South Korea	HELLP syndrome
Liu S. et al., 2011	Canada	Eclampsia
**Hemorrhagic disorders**		
Bhandari S. et al., 2014	UK	Antepartum hemorrhage
McCormack et al., 2008	Australia	Antepartum hemorrhage
Yang and Savitz et al., 2001	USA	Antepartum hemorrhage
Baldwin H. et al., 2017	Australia	Hemorrhagic AIP
Jakobsson M. et al., 2015	Finland	Hemorrhagic AIP
		Hysterectomy
Patel E. et al., 2015	USA	Postpartum hemorrhage
		Transfusion
Sheiner E. et al., 2005	Israel	Postpartum hemorrhage
Jakobsson M. et al., 2015	Finland	Uterine rupture
Kaczmarczyk M. et al., 2007	Sweden	Uterine rupture
Ofir K. et al., 2003	Israel	Uterine rupture
Ronel D. et al., 2012	Israel	Uterine rupture
Vilchez G. et al., 2017	USA	Uterine rupture
**Cardiovascular disorders**		
Patel E. et al., 2015	USA	Acute heart failure
		Myocardial infarction/Ischemia
Henry D. et al., 2016	USA	Arrhythmia
Kao D. et al., 2013	USA	Peripartum myocardiopathy
Aarnio K. et al., 2017	Finland	Stroke
Kang J. et al., 2010	Taiwan	Stroke
**Hepatic disorders**		
Brouwers L. et al., 2015	Netherlands	Severe intrahepatic cholestasis
Geenes V. et al., 2014	UK	Severe intrahepatic cholestasis
Herrera C. et al., 2018	USA	Severe intrahepatic cholestasis
Kawakita T. et al., 2015	USA	intrahepatic cholestasis
Rioseco A.J. et al. 1994	USA	Intrahepatic cholestasis
Wikstrom S.E. et al., 2013	Sweden	Intrahepatic cholestasis
**Renal discords**		
Hildebrand A. et al., 2015	Canada	Acute kidney injury
Patel E. et al., 2015	USA	Acute renal failure
**Thromboembolic disorders**		
Kramer M. et al., 2012	Canada	Amniotic fluid embolism
Kramer M. et al., 2013	USA	Amniotic fluid embolism
Roberts C. et al., 2010	Australia	Amniotic fluid embolism
Spiliopoulos et al.2009	USA	Amniotic fluid embolism
Ben-Joseph R. et al., 2009	Israel	Deep venous thrombosis
Patel E. et al., 2015	USA	Deep venous thrombosis
		Pulmonary edema
		Pulmonary embolism
Morris J. et al., 2010	Australia	Pulmonary embolism

SMM—severe maternal morbidity, HELLP—haemolysis, elevated liver enzymes, low platelet, AIP—abnormally invasive placenta, USA—United States of America, UK—United Kingdom.

**Table 2 jcm-09-02035-t002:** Detailed characteristics of included studies and quality rating.

Author (Year)	Country	Study Type/Design	Data Source/Setting	Study Population	Participants	SMM Definition	Adverse Perinatal Outcomes	Key Findings (Effect of SMM on Respective Perinatal Outcomes)	Quality Score (Rating)
Aarnio K. et al., 2017 [48]	Finland	Retrospective cohort study (matched) ^a^	Linked data (Medical Birth registry and Helsinki Young Stroke Registry)	Pregnant women	760 women	Ischemic stroke	Perinatal death	↔ IRR 5.43, (95% CI: 0.80–37.00)	7 (Good)
							Small for gestational age	↔ IRR 2.01, (95% CI: 0.87–4.64)	
							Low birth weight	↔ IRR 1.37, (95% CI: 0.79–2.36)	
							5-min Apgarscore < 7	↔ IRR 0.98, (95% CI: 0.33–2.97)	
Alsulyman O.M. et al. 1996 [22]	USA	Retrospective cohort study	Medical records over 7 years	Women who had antepartum intrahepatic cholestasis of pregnancy	158 patients	Intrahepatic cholestasis of pregnancy	Preterm birth	↑ 2-fold (14% of cases vs. 7.6% of controls)	5 (Fair)
							Small for gestational age	↑ 2-fold (7.6% for cases vs. 3.8 for controls)	
							Stillbirth/fetal death	2 cases vs. 0 for controls)	
Baldwin H. et al., 2017 [23]	Australia	Retrospective cohort study	Linked data ^b^	Women delivered a live born or stillborn infant(s) (>20 weeks of gestation)	922,925 deliveries	Hemorrhagic abnormally invasive placenta	Preterm birth	Preterm is higher in AIP (25.5% vs. 7.4)↑RR 5.8, (95% CI: 4.9–7.0) for < 32 weeks↑ RR 3.2, (95% CI: 2.8–3.8) for 33-36 weeks)	8 (Good)
							Neonatal death	↑ ARR 3.1, (99% CI: 2.7–3.5)	
							Stillbirth/fetal death	↑ RR 5.4, (99% CI: 4.0-7.3)	
							5-min Apgarscore < 7	↔ RR 1.3, (99% CI: 0.84–2.077) ^d^	
							Small for gestational age	↑ RR 1.24, (99% CI: 1.10–1.40) ^d^	
							NICU admission	↑ RR 1.12, (99% CI: 1.27–5.44) ^d^	
Ben-Joseph R. et al., 2009 [24]	Israel	Population-based cohort study	Hospital data	All pregnant women with and without a history of deep venous thrombosis	212,086 deliveries	Deep venous thrombosis	Preterm birth	↑ AOR 1.8, (95% CI: 1.1-2.9)	7 (Good)
							5-min Apgarscore < 7	↔ OR 1.31 (95% CI: 0.18-9.41)	
							Perinatal death	↔ OR 1.65, (95% CI: 0.52-5.20)	
Bhandari S. et al., 2014 [25]	UK	Population-based cohort study	Hospital data	All women who gave singleton birth (≥24 weeks of gestation)	75,940 women	Abnormal bleeding of unknown origin	Preterm birth	↑ AOR 2.30, (95% CI: 2.11-2.50)	8 (Good)
							Stillbirth/fetal death	↔ AOR 0.92, (95% CI: 0.66-1.30)	
							Neonatal death	↔ AOR 0.92 (95% CI: 0.61–1.38)	
							Low birth weight (<2500 g)	↔ AOR 0.90, (95% CI: 0.79–1.03)	
							NICU admission	↔ AOR 1.03: 95% CI: 0.94–1.12)	
Brouwers L. et al., 2015 [26]	Netherlands	Retrospective cohort study	Hospital data	Women with singleton pregnancies and diagnosed with intrahepatic cholestasis of pregnancy	215 women	Intrahepatic cholestasis of pregnancy	Small for gestational age	↔ OR 1.98, (95% CI: 0.89–4.43) ^f^	7 (Good)
							Preterm birth	Higher in ICP cases (19.3% vs. 6.8%)	
							NICU admission	↔ OR 1.91, (95% CI: 0.54–6.74) ^f^	
							Perinatal death	↔ OR 1.88, (95% CI: 0.26–13.56) ^f^ ↑ AOR 1.26, (95% CI: 1.01–1.57)per 10 micro mol/liter	
Buchbinder A. et al., 2002 [27]	USA	Prospective cohort study	Hospital data	Women who had preeclampsia for their first birth	598 women	Severe gestational hypertension	Preterm birth	↑ OR 7.18, (95% CI: 4.21–12.25) ^f^	7 (Good)
							Small for gestational age	↑ OR 2.55, (95% CI: 1.19–5.43) ^f^	
							Stillbirth/fetal death	↔ OR 2.96, (95% CI: 0.77–11.43) ^f^	
							Neonatal death	↔ OR 1.93, (95% CI: 0.21–17.52) ^f^	
							NICU admission	↑ OR 2.78, (95% CI: 1.57–4.91) ^f^	
							Intraventricular hemorrhage	↔ OR 1.08, (95% CI: 0.06–21.17) ^f^	
Carte E. et al., 2017 [28]	USA	Retrospective cohort study	Hospital data	Women admitted to labor and delivery unit and gave live birth	1905 women	Severe preeclampsia	5-min Apgarscore < 7	↑ AOR 2.40, (5% CI: 1.58–3.65)	7 (Good)
							NICU Admission	↑ AOR 3.38, (95% CI: 2.45–4.67)	
							Neonatal death	↔ AOR 0.71, (95% CI: 0.35–1.42)	
							Adverse neonatal outcome	↑ AOR 3.66, (95% CI: 2.71–4.93)	
Geenes V. et al., 2014 [49]	UK	Case-control study	UK Obstetric Surveillance System (UKOSS)	Women with intrahepatic cholestasis of pregnancy	669 women	Intrahepatic cholestasis of pregnancy	Preterm birth	↑ AOR 5.39, (95% CI: 4.17–6.98)	7 (Good)
							Small for gestational age	↓ RR 0.70, (95% CI: 0.54–0.91)	
							Stillbirth/fetal death	↑ AOR 2.58, (95% CI: 1.03–6.49)	
							5-min Apgarscore < 7	↔ AOR 1.92, (95% CI: 0.92–3.99)	
							NICU admission	↑ AOR 2.68, (95% CI: 1.97–3.65)	
Haddad B. et al., 2000 [50]	USA	Case-control study	Hospital data	Women with preeclampsia or without the HELLP syndrome	64 women	HELLP syndrome	Small for gestational age	↑ OR 3.4, (95% CI: 1.0–11.3)	8 (Good)
							Intraventricular hemorrhage	↔ OR 0.5, (95% CI: 0.0–6.1)	
							Neonatal death	↔ OR 0.6, (95% CI: 0.1–2.8)	
Henry D. et al., 2016 [29]	USA	Retrospective cohort study	Hospital data	Pregnant women with cardiac diseases	143 women	Cardiac arrhythmia	Stillbirth/fetal death	↔ OR 0.0, (95% CI: 0.0–5.78)	5 (Fair)
							Preterm birth	↔ OR 0.79, (95% CI: 0.32–1.99)	
							Intrauterine growth restricts	↑ OR 4.08, (95% CI: 1.23–13.54)	
							Small for gestational age	↔ OR 0.47, (95% CI: 0.10-2.19)	
							5-min Apgarscore < 7	↔ OR 1.58, (95% CI: 0.57–4.46))	
							NICU admission	↔ OR 0.86, (95% CI: 0.38–1.97)	
Herrera C. et al., 2018 [30]	USA	Retrospective cohort study	Administrative and clinical electronic data (from 22 hospital) administered by Intermountain Healthcare System	Women with intrahepatic cholestasis of pregnancy based on serum bile acid test	785 mothers	Severe intrahepatic cholestasis of pregnancy	Adverse neonatal outcome	↑ ARR 5.6, (95% CI: 1.3–23.5)	6 (Fair)
							Small for gestational age	↔ ARR 2.19, (95% CI: 0.79–6.05)	
							NICU admission	↔ ARR 0.91, (95% CI: 0.48–1.74)	
Hildebrand A. et al., 2015 [31]	Canada	Retrospectivecohort study	Linked health care databases (seven databases)	Women with acute kidney injury treated with dialysis during pregnancy or postpartum period (≥20 weeks of gestation)	1,918,789 deliveries	Acute kidney injury	Low birth weight	↑ RR, 4.66, (95% CI: 3.64–5.96)	6 (Fair)
							Small for gestational age	↑ RR 3.16, (95% CI: 1.90–5.27)	
							Preterm birth	↑ RR 2.49, (95% CI: 2.03–3.06)	
							Stillbirth/fetal death	There were zero cases in AKI and 0.1% in non-AKI group	
							Neonatal death	↑ risk (2.7% vs. 0.8)	
							Perinatal mortality	↑ risk (2.7% vs. 0.4)	
							Adverse perinatal outcomes	↑ RR 3.40, (95% CI: 2.52–4.58)	
Jakobsson M. et al., 2015 [51]	Finland	Case-control study	Nordic Obstetric Surveillance Study (NOSS)	Cases: women developed obstetric near-miss events (uterine rupture, abnormally invasive placenta, and emergency peripartum hysterectomy). Controls: all other births	145,743 women	Hemorrhagic abnormally invasive placenta	5-min Apgarscore < 7	↑ AOR 3.46, (95% CI:1.37–8.73)	7 (Good)
							Perinatal death	↑ AOR 5.40, (95% CI: 1.30–22.5)	
							Stillbirth/fetal death	↔ OR 5.42, (95% CI: 0.77–38.0)	
							Preterm birth	↑ OR 7.72, (95% CI: 5.82–10.2	
							Neonatal death	↑OR 9.87, (95% CI: 1.41–69.2)	
							NICU admission	↑ AOR 2.75, (95% CI:1.54–4.91)	
							Low birth weight (<2500 g)	↑ AOR 8.30, (95% CI: 4.52–15.2)	
						Hysterectomy	5-min Apgarscore < 7	↑ AOR 3.75, (95% CI: 1.28–11.0)	
							Preterm birth	↑OR 5.99, (95% CI: 3.71–9.68)	
							NICU admission	↑ AOR 11.8, (95% CI: 9.0–15.6)	
							Low birth weight (<2500 g)	↔ AOR 1.74, (95% CI: 0.67–4.53)	
							Stillbirth/fetal death	↑ OR 10.3, (95% CI: 1.50–71.3)	
							Neonatal death	↔ OR 9.2, (95% CI: 0.56–151.2)	
						Uterine rupture	Perinatal death	↑ AOR 11.8, (95% CI: 5.39–25.8)	
							Preterm birth	↔ OR 1.33 (95% CI: 0.71–2.49)	
							NICU admission	↑ AOR 1.98 95% CI: 1.28–3.04)	
							Neonatal death	↑ OR 10.2, (95% CI: 2.57–40.6)	
							Stillbirth/fetal death	↑ OR 16.8, (95% CI: 7.67–36.9)	
							Low birth weight (<2500 g)	↔ AOR 1.29, (95% CI: 0.62–2.66)	
							5-min Apgarscore < 7	↑ AOR 10.5, (95% CI: 6.82–16.3)	
Kaczmarczyk M. et al., 2007 [32]	Sweden	Prospective cohort study	Swedish Birth Register (population based)	Women with live single births	300,200 women	Uterine rupture	Low birth weight (<2500 g)	↔ AOR 0.58, (95% CI: 0.31–1.08)	6 (Fair)
							Preterm birth	↔ AOR 0.34, (95% CI: 0.08–1.45)	
							Neonatal death	↑ AOR 65.62, (95% CI: 32.60–132.08)	
Kang J. et al., 2010 [52]	Taiwan	Case-control study	Linked population-based datasets (Taiwan National Health Insurance Research Dataset (NHIRD) and the national birth certificate registry)	Cases: women who have stroke during their pregnancy period Controls: randomly selected women with no history of stroke	1,449 women	Stroke	Preterm birth	↔ AOR 0.72, (95% CI: 0.35 to 1.50)	7 (Good)
							Low birth weight (<2500 g)	↔ AOR 0.75, (95% CI: 0.36 to 1.54)	
							Small for gestational age	↔ AOR 0.84, (95% CI: 0.52 to 1.36)	
Kao D. et al., 2013 [33]	USA	Retrospective cohort study	Hospital data	Women who were admitted for delivery	4,003,914 women	Peripartum cardiomyopathy	Stillbirth/fetal death	↑ OR 3.74, (95% CI: 1.69–5.64)	6 (Fair)
Kawakita T. et al., 2015 [34]	USA	Retrospective cohort study (multicentre)	Hospital data	All women diagnosed with intrahepatic cholestasis of pregnancy	233 women	Intrahepatic cholestasis of pregnancy	Adverse perinatal outcome	↔ AOR 4.28, (95% CI: 0.71–25.83)	8 (Good)
							Preterm birth	↑ OR 2.49, (95% CI: 1.36–4.57) ^f^	
							Stillbirth/fetal death	↔ OR 17.71, (0.94–333.16) ^f^	
							NICU admission	↔ OR 1.37, (95% CI: 0.72–2.58) ^f^	
							Low birth weight	↑ OR 2.44, (95% CI: 1.07–5.56) ^f^	
							Intrauterine growth restricts	↑ OR 1.66, (95% CI: 0.54–5.11) ^f^	
Kim H. et al., 2006 [53]	South Korea	Matched case control study^c^	Hospital data	Women with singleton pregnancy and complicated by HELLP syndrome	121 pregnancies	HELLP syndrome	Neonatal death	↑ OR 11.5, (95% CI: 1.2–110.4)	4 (Poor)
							Intraventricular hemorrhage	↑ OR 39.0, (95% CI: 7.4–206.4)	
							NICU admission	↑ OR 19.0, (95% CI: 4.8–75.8)	
							5-min Apgarscore < 6 *	↔ OR 0.4, (95% CI: 0.1–1.2)	
Kramer M. et al., 2012 [36]	Canada	Retrospective cohort study	Hospital data collected by Canadian Institute for Health Information (CIHI)	All women and deliveries registered in Canadian Institute of Health Information (CIHI) database	4,508,462 deliveries	Amniotic fluid embolism	Stillbirth	↑ AOR 5.9, (95% CI: 2.0–17.4)	7 (Good)
							Intrauterine growth restricts	↑ AOR 1.6, (95% CI: 0.7–3.5)	
							Low birth weight	↑ AOR 1.8, (95% CI: 1.8–1.8)	
Kramer M. et al., 2013 [35]	USA	Population-based cohort study	Hospital data (US Nationwide Inpatient Sample)	Women with amniotic fluid embolism	8,571,209 deliveries	Amniotic fluid embolism	Stillbirth	↔ AOR 2.1, (95% CI: 0.8–5.5)	8 (Good)
Liu S. et al., 2011 [37]	Canada	Population-based cohort study	Hospital data	Women with eclampsia and their respective deliveries	1,910,729 women	Eclampsia	Small for gestational age	↑ AOR 2.6, (95% CI: 2.3–3.0)	8 (Good)
							Preterm birth	↑ AOR 4.808, (95% CI: 4.330–5.338) ^e^	
							NICU admission	↑ AOR 2.8, (95% CI: 2.4–3.2)	
							Stillbirth/fetal death	↑ AOR 2.4, (95% CI: 1.5–3.9)	
							Neonatal death	↑ AOR 2.9, (95% CI: 1.6–5.5)	
McCormack et al., 2008 [38]	Australia	Retrospective cohort study	Hospital data	Women with singleton deliveries	28,014 deliveries	Abnormal bleeding of unknown origin	Preterm birth	↑ AOR 4.31, (95% CI: 3.84–4.84)	7 (Good)
							Stillbirth/fetal death	Stillbirth is not associated with ABUO and none-ABUO cases (0.90% vs. 0.95%	
							5-min Apgar < 7	↔ AOR 1.05, (95% CI: 0.76–1.44)	
							NICU admission	↑ AOR 1.23, (95% CI: 1.01–1.51)	
							Neonatal death	ABUO is associated with early neonatal death (1.3 versus 0.3%)	
							Perinatal death	↔ AOR 0.67, (95% CI: 0.43–1.08)	
McPherson J. et al., 2013 [39]	USA	Retrospective cohort study	Hospital data	Women who have singleton, non-anomalous pregnancies with complete outcome data	47,118 women	Seizure disorder	Intrauterine growth restricts	↔ AOR 1.11, (95% CI: 0.82–1.50)	7 (Good)
							Intrauterine growth restricts	↔ AOR 0.82, (95% CI: 0.56–1.20)	
							Stillbirth	↑ OR 1.70, (95% CI: 0.55–5.28)	
							Preterm birth	↔ AOR 1.06, (95% CI: 0.81–1.38)	
Morris J. et al., 2010 [40]	Australia	Retrospective cohort study	Linked dataset	Women who had deliveries ≥ 20 weeks of gestation	380,459 women	Pulmonary embolism	Stillbirth	↑ AOR 5.97, (95% CI: 3.09-11.6)	8 (Good)
							Preterm birth	↑ AOR 2.18, (95% CI: 1.54–3.09)	
							Small for gestational age	↔ AOR 1.23, (95% CI: 0.84–1.81)	
Ofir K. et al., 2003 [41]	Israel	Population based cohort study	Hospital data	All women with singleton pregnancy and delivered with and without uterine rupture	117,685 women	Uterine rupture	Perinatal death	↑ OR 17.2, (95% CI: 7.3–38.7)	4 (poor)
							Low birth weight	↑ OR 1.21, (95% CI: 0.43–3.39) ^f^	
							5-min Apgarscore < 5	↑ OR 42.8, (95% CI: 12.8–126.8) ^f^	
Patel E. et al., 2015 [56]	USA	Cross-sectional study (Prospective)	Hospital data (>1000 hospitals)	All delivery records containing of women with stillbirth	12,524,119 deliveries	Cardiac arrest	Stillbirth	↑ OR 14.84, (95% CI: 10.97–20.07)	8 (Good)
						Pulmonary edema	Stillbirth	↑ OR 7.66, (95% CI: 5.94–9.89)	
						Acute respiratory distress	Stillbirth	↑ OR 12.25, (95% CI: 10.30–14.57)	
						Pulmonary embolism	Stillbirth	↑ OR 5.06, (95% CI: 4.00–6.42)	
						Deep venous thrombosis	Stillbirth	↑ OR 2.89, (95% CI: 2.29–3.64)	
						Sepsis	Stillbirth	↑ OR 12.29, (95% CI: 10.94–13.80)	
						Acute renal failure	Stillbirth	↑ OR 20.00, (95% CI: 18.28–21.88)	
						Postpartum hemorrhage	Stillbirth	↑ OR 1.65, (95% CI: 1.58–1.72)	
						Chorioamnionitis	Stillbirth	↑ OR 2.74, (95% CI: 2.65–2.84)	
Rioseco, A.J. et al. 1994 [54]	USA	Case control study	Medical record	Women with intrahepatic cholestasis	640 patients	Intrahepatic cholestasis of pregnancy	Preterm birth	↑ 3-fold, (19.3% vs. 6.8%)	6 (Fair)
							5-min Apgarscore < 7	↑ 1.3-fold, (2.2% vs. 1.3%)	
							Small for gestational age	↑ 1.4-fold, (6.3% vs. 4.4%)	
							Stillbirths	Higher in ICP cohort (12 vs.9 per 1000 births)	
							Neonatal deaths	Higher in ICP cohort (6 vs. 3 per 1000 births)	
							Perinatal death	Higher in ICP cohort (18 vs.612 per 1000 births)	
Roberts C. et al., 2010 [42]	Australia	Population-based cohort study	Linked data (birth, hospital, and death data)	All women and deliveries	606,393 deliveries	Amniotic fluid embolism	Preterm birth	↔ RR 1.9, (95% CI: 0.4–8.6)	6 (Fair)
							Perinatal death	Perinatal death rate was 32% (95% CI: 12–56)	
							Stillbirth/fetal death	Stillbirth was higher in AFE group (26% vs. 0%)	
Ronel D. et al., 2012 [43]	Israel	Population-based cohort study	Perinatal database	All singleton births	240,189 deliveries	Uterine rupture	Preterm birth	↑ AOR 2.48, (95% CI: 1.49–4.12)	6 (Fair)
							5-min ApgarScore < 5 *	↑ OR 9.59, (95% CI: 6.45–14.24)	
							Perinatal death	↑ AOR 17.4, (95% CI: 9.87–23.88)	
Sheiner E. et al., 2005 [44]	Israel	Population-based cohort study	Hospital data	Deliveries complicated by postpartum hemorrhage	154,311 deliveries	Postpartum hemorrhage	Small for gestational age	↔ OR 1.19, (95% CI: 0.89–1.58) ^f^	7 (Good)
							5-min Apgarscore < 7	↔ OR 0.75, (95% CI: 0.24–2.330) ^f^	
							Preterm birth	↔ OR 1.51, (95% CI: 0.89-2.57) ^f^	
							Low birth weight	↔ OR 0.94, (95% CI: 0.72, 1.23) ^f^	
							Perinatal death	↑ by 3.5%	
Spiliopoulos et al., 2009 [45]	USA	Population-based cohort study	Perinatal linked data set	All births from 1997 to 2005	1,004,116 births	Amniotic fluid embolism	NICU admission	6-fold, (48.6% vs. 8.1%)	6 (Fair)
							5-min Apgarscore < 7	Low Apgar score is higher in AFE cases (22.2% vs. 0.5.6%	
Vilchez G. et al., 2017 [55]	USA	Cases-control study (Prospective)	CDC and National Centre for Health Statistics (NCHS) birth database	Cases: women with uterine rupture. Controls: Women with no uterine rupture	5690 women	Uterine rupture	NICU admission	↑ AOR 3.88, (95% CI: 3.28–4.60)	7 (Good)
							Low birth weight	↑ OR 9.2, (95% CI: 7.2–11.6)	
Wikstrom Shemer E. et al., 2013 [46]	Sweden	Population-based cohort study	Linked data (Hospital data plus Swedish Medical Birth Register (MBR))	Women with singleton deliveries	1,213,668 deliveries	Intrahepatic cholestasis of pregnancy	Preterm birth	↑ OR 2.93, (95% CI: 2.71–3.17) ^f^	6 (Fair)
							Stillbirth	↔ AOR 0.92, (95% CI: 0.52–1.62)	
							5-min Apgarscore < 7	↑ AOR 1.45, (95% CI: 1.14–1.85)	
							Neonatal death	↔ AOR 0.45, (95% CI: 0.15–1.40)	
							Small for gestational age	↓ AOR 0.44, (95% CI: 0.32–0.60)	
Yang and Savitz 2001 [47]	USA	Population-based cohort study	US Maternal and Infant Health Survey	Women with vaginal bleeding during pregnancy	9953 births	Antepartum hemorrhage	Preterm birth	↑ OR 2.81, (95% CI: 2.48–3.18) ^f^	4 (Poor)
							Small for gestational age	↑ OR 1.25, (95% CI: 1.07–1.46) ^f^	

^a^ Matched by maternal age, year of delivery, parity, residence and number of newborns; ^b^ Data link is from NSW Register of Births, Deaths and Marriages (2003–2013) (death data), NSW Perinatal Death Review Database (2003–2009) (stillbirth data) and classification resources; ^c^ Samples were matched by gestational age, race, infant gender, and mode of delivery; ^d^ pooled estimates (using fixed effect model) from stratified data presented for pre-term and term births neonates; ^e^ pooled estimate (fixed effect model) from stratified; data presented for very preterm (22–31 weeks) and mild preterm (32–36 weeks) estimates; ^f^ effect estimate computed from available data in the study; ↑ Significant positive association; ↓ Significant negative association; ↔ No significant association. Abbreviations: RR—relative risk, IRR—incidence risk ratio, AIP—abnormally invasive placenta, HELLP—hemolysis, elevated liver enzymes, low platelet, OR—odds ratio, AOR—adjusted odds ratio, NICU—neonatal intensive care unit, USA—United States of America, UK—United Kingdom.

**Table 3 jcm-09-02035-t003:** Adverse perinatal outcomes, number of reports from studies, effect estimate, citations, and heterogeneity.

	Effect Estimate (Odds Ratio (M-H, Random, 95% CI))	Citations	Heterogeneity (I^2^), %
**Preterm birth**	**3.11 (2.56–3.78)**	[22,23,24,25,26,27,29,31,32,34,37,38,40,42,46,47,49,51,52,54]	**95**
Hemorrhagic disorders	3.42 (2.55–4.58)	[23,25,32,38,47,51]	96
Hypertensive disorders	6.79 (6.06–7.60)	[27,37]	0
Cardiovascular disorders	0.78 (0.44–1.37)	[29,52]	0
Hepatic disorders	3.19 (2.46–4.13)	[22,26,34,46,49,54]	64
Thromboembolic disorders	2.40 (1.67–3.46)	[24,40,42]	30
Acute kidney disorders	3.31 (2.44–4.50	[31]	NA
Test for sub-group differences			**95.5** (X^2^ = 110.30, *p* < 0.0001)
**Small for gestational age (SGA)**	**1.33 (0.98**–**1.81)**	[22,23,24,26,27,29,30,31,37,40,44,46,47,48,49,50,52,54]	**93**
Hemorrhagic disorders	1.09 (0.83–1.42)	[23,44,47]	83
Hypertensive disorders	2.86 (2.51–3.25)	[27,37,50]	0
Cardiovascular disorders	1.01 (0.53–1.90)	[29,48,52]	53
Hepatic disorders	0.95 (0.51, 1.77)	[22,26,30,46,49,54]	86
Acute renal disorders	3.52 (2.08–5.97)	[31]	NA
Thromboembolic disorders	1.48 (1.09–1.99)	[24,40]	0
Test for sub-group differences			**92.3** (X^2^ = 65.20, *p* < 0.0001)
**Low birth weight (<2500 g)**	**2.20 (1.56**–**3.09)**	[23,31,34,38,41,43,44,48,51,52,55]	**94**
Hemorrhagic disorders	2.31 (1.57–3.40)	[23,38,41,43,44,51,55]	95
Cardiovascular disorders	0.91 (0.61–1.38)	[48,52]	0
Acute renal disorders	6.39 (4.62–8.83)	[31]	NA
Hepatic disorders	2.44 (1.07–5.56)	[34]	NA
Test for sub-group differences			**94.5** (X^2^ = 54.50, *p* < 0.0001)
**5-min Apgar score < 7**	**3.66 (2.41**–**5****.56)**	[23,24,28,29,38,41,43,44,45,46,48,49,51,53,54]	**93**
Hemorrhagic disorders	4.16 (2.54–6.81)	[23,38,41,43,44,51]	92
Hypertensive disorders	4.61 (1.17–18.20)	[28,53]	80
Cardiovascular disorders	1.26 (0.63–2.52)	[29,48]	0
Hepatic disorders	1.97 (0.95–4.05)	[46,49,54]	74
Thromboembolic disorders	8.93 (0.07–1086.45)	[24,45]	95
Test for sub-group differences			56.5 (X^2^ = 9.20, *p* = 0.06)
**NICU admission**	**3.22 (2.45**–**4.25)**	[23,25,26,27,28,29,34,37,38,45,49,51,53,55]	**97**
Hemorrhagic disorders	3.34 (2.26–4.94)	[23,25,38,51,55]	98
Hypertensive disorders	3.63 (2.63–5.02)	[27,28,37,53]	64
Cardiovascular disorders	0.86 (0.37–1.99)	[29]	NA
Hepatic disorders	1.89 (1.11–3.20)	[26,34,49]	45
Thromboembolic disorders	10.81 (6.02–19.39)	[45]	NA
Test for sub-group differences			86.8 (X^2^ = 30.30, *p* < 0.0001)
**Stillbirth**	**4.87 (2.63**–**6****9.01)**	[22,23,25,27,29,31,33,34,37,38,40,42,46,49,51,53,54,56]	**99**
Hemorrhagic disorders	3.40 (1.88–6.15)	[23,25,38,51,56]	96
Hypertensive disorders	2.74 (1.73–4.34)	[27,37,53]	0
Cardiovascular disorders	15.24 (1.29–180.60)	[29,33,56]	98
Hepatic disorders	1.95 (0.82–4.67)	[22,34,46,49,54]	58
Acute renal disorders	15.16 (4.41–52.14)	[31,56]	32
Thromboembolic disorders	5.07 (3.12–8.24)	[40,42,56]	91
Test for sub-group differences			**68.3**(X^2^ = 15.70, *p* < 0.008)
**Neonatal death**	**4.02 (2.45**–**6.59)**	[23,25,27,28,31,32,37,38,46,50,51,53,54]	**89**
Hemorrhagic disorders	7.33 (3.06–17.53)	[23,25,32,38,51]	94
Hypertensive disorders	3.00 (1.78–5.07)	[27,28,37,50,53]	39
Hepatic disorders	0.92 [0.47–1.79]	[46,54]	0
Acute renal disorders	3.28 (1.35–7.97)	[31]	NA
Test for sub-group differences			**80.4** (X^2^ = 15.30, *p* = 0.002)
**Perinatal death**	**4.74 (2.47**–**9.12)**	[24,26,31,38,41,43,44,48,51,54]	**89**
Hemorrhagic disorders	6.18 (2.55–14.96)	[38,41,43,44,51]	93
Cardiovascular disorders	3.92 (0.79–19.57)	[48]	NA
Hepatic disorders	1.63 (0.81–3.28)	[26,54]	0
Acute renal disorders	7.13 (2.93–17.35)	[31]	NA
Thromboembolic disorders	1.66 (0.53–5.21)	[24]	NA
Test for sub-group differences			**60.6** (X^2^ = 10.20, *p* = 0.04)

Funnel plots for each adverse perinatal outcome to assess the effect of publication bias are presented in Appendix A. Sensitivity analyses were also performed for each adverse outcome after the exclusion of low-quality studies [22,29,41,47,53] but they did not affect the significance of any of the outcomes of interest.

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
