# Peer review of "The Impact of Severe Maternal Morbidity on Perinatal Outcomes in High Income Countries: Systematic Review and Meta-Analysis"

_jcm, 2020, doi:10.3390/jcm9072035_

Round 1
Reviewer 1 Report
In general, this is an interesting paper about a very important topic. The methodology seems robust and I only have a few minor comments.
The introduction is relevant and well-written.
Line 59: Would be good to specify the reasons for increased prevalence (obesity, unhealthy diets?)
Line 71: put the reference before the comma.
The methods section reads well and from what I can see, includes nearly all relevant segments.
Line 111: Is there a reference for the cut-off for NOS-score or was the cut-off chosen particularly for this study?
Line 133: could more detail be added on how the sensitivity analysis was performed?
Results:
Figure 1: check size of boxes so that all print is visible.
Table 1: please explain abbreviations below the table
Table 2: widen column 2. Typo Brouwers 2015: mmol/liter. Typo Hildebrand 0 0.1%. Explain abbreviations AIP, HELLP and IRR below table.
Typo line 166: tern instead of term, “term birth neonates” is correct spelling?
Line 201: Why only women with severe hemorrhage? From the forest plot it seems that renal disorders and hepatic disorders are also significantly associated with the outcome? If there is a general rule that topics with only one relevant paper should not be reported, maybe this could be expressed somewhere (maybe not all readers are aware?)
Line 223: change to “SMM and stillbirth”
Discussion:
Line 250: Change to “We found that women who experienced…”
Line 253: Would be interesting to know some more detail about which diagnoses the heterogeneity of definitions was largest and by how much it varied.
Line 260: this reads as if it could be an example of reverse causality, but probably not intended?
Line 263: maybe specify through which proposed mechanisms? Epigenetics?
Line 266: could you focus here on primary prevention in addition to prevention of adverse outcomes?
Line 273: “associated with a variety of…”
Line 274: Suggest adding: “This knowledge should be of benefit for clinicians”
Line 276: repetitive language
Should not a high heterogeneity (I2) for total effect be expected when varied determinants are included for each outcome? This should be mentioned in the discussion, since high heterogeneity is mentioned as a main finding throughout the paper.
Line 289: would you care to elaborate?
Line 300: add “some of the funnel plots…”
Author Response
Dear reviewer,
We would like to thank you for your thoughtful comments and efforts towards improving our manuscript. In the file attached, we highlight all concerns of reviewers and provided point by point response each comments and suggestions.
Kindest regards
Sailesh Kumar
Reviewers comments and suggestions
- Line 59: Would be good to specify the reasons for increased prevalence (obesity, unhealthy diets?)
Response: Comment accepted. Furthermore, although precipitous declines in maternal mortality rates have occurred in many countries, rates of severe maternal morbidity (SMM) have not shown similar improvements and are increasing in some regions mainly because of high rates of pre-existing co-morbidities including diabetes and obesity [2]. [Line 62-63]
- Line 71: put the reference before the comma.
Response: Done [Line 71]
- Line 111: Is there a reference for the cut-off for NOS-score or was the cut-off chosen particularly for this study?
Response: Reference [20] is the appropriate reference. We have moved this to the end of the sentence in line 112.
- Line 133: could more detail be added on how the sensitivity analysis was performed?
Response: We have revised this sentence to read: Sensitivity analyses were performed by sequentially removing studies at high risk of bias to evaluate the impact of SMM on different adverse perinatal outcomes.
- Figure 1: check size of boxes so that all print is visible.
Response: The size of the boxes made to fit texts [Figure 1]
- Table 1: please explain abbreviations below the table
Response: The relevant abbreviations have now been detailed in the table legend.
- Table 2: widen column 2. Typo Brouwers 2015: mmol/liter. Typo Hildebrand 0 0.1%. Explain abbreviations AIP, HELLP and IRR below table.
Response: The columns have been widened and the relevant abbreviations have now been detailed in the table legend.
- Typo line 166: tern instead of term, “term birth neonates” is correct spelling?
Response: This has been corrected
- Line 201: Why only women with severe haemorrhage? From the forest plot it seems that renal disorders and hepatic disorders are also significantly associated with the outcome? If there is a general rule that topics with only one relevant paper should not be reported, maybe this could be expressed somewhere (maybe not all readers are aware?)
Response: These associations were reported only in 1 paper thus rendering it not possible to produce pooled estimates.
- Line 223: change to “SMM and stillbirth”
Response: Change made.
Discussion:
- Line 250: Change to “We found that women who experienced…”
Response: Change made. [Line 263]
- Line 253: Would be interesting to know some more detail about which diagnoses the heterogeneity of definitions was largest and by how much it varied.
Response: Because of inconsistent definitions for SMM we were unable to compare the prevalence of SMM in HICs nor the magnitude of heterogeneity for the various definitions.
- Line 260: this reads as if it could be an example of reverse causality, but probably not intended?
Response: We have rephrased this sentence to avoid the inference of reverse causality.
- Line 263: maybe specify through which proposed mechanisms? Epigenetics?
Response: As there is no good evidence in the literature to suggest a causal pathway, we have avoided speculating on possible causes.
- Line 266: could you focus here on primary prevention in addition to prevention of adverse outcomes?
Response: The sentence now reads as: …… highlighting the crucial importance of primary prevention and reducing the burden of SMM to mitigating the burden of SNM [2] [60] and its associated consequences [10, 12]. It is an imperative that transcends the socioeconomic status of a country [Line 298-300].
- Line 274: Suggest adding: “This knowledge should be of benefit for clinicians”
Response: We have made this change.
- Should not a high heterogeneity (I2) for total effect be expected when varied determinants are included for each outcome? This should be mentioned in the discussion, since high heterogeneity is mentioned as a main finding throughout the paper.
Response: We have mentioned methodological issues that could have introduced high heterogeneity in the “strengths and limitation section” and reads as: …..the use of heterogeneous SMM definitions, differences in participant characteristics and sampling procedures would also have introduced heterogeneity into our analysis. However, our use of random‐effects modeling [76] as well as the use of components within a widely accepted and standard SMM definition mitigates this limitation [Line 324-328].
- Line 289: would you care to elaborate?
Response: We have added the following sentence: One of the limitations of this review is the use of the WHO-near miss criteria to define SMM as these criteria are not consistently used by all HICs.
- Introduction Lines 74-83: Suggest emphasizing the costs of SMM in HICs to further justify why HICs were the focus of this study. Much attention in the MCH realm understandably goes to LMICs because of immediate need; however, the problem of SMM is still serious, even if it is not typically resulting in maternal mortality. Providing some information about the consequences of SMM to the healthcare system, economy, quality of life, etc. would be helpful to contextualize this issue.
Response: we thank the reviewer for this comment. In this systematic review we only aimed to show how the association between SMM and perinatal outcomes in HICs. In the introduction section [line 69 – 73] we showed that HICs also have high burden of SMM by giving different data driven evidence. Hence, we limited our focus of introduction and discussion points in the continuum of SMM and adverse perinatal outcomes rather than presenting the economic, healthcare and social consequences of SMM which is not the focus of this paper.

Reviewer 2 Report
This is a well written comprehensive manuscript looking at severe maternal morbidity's (SMM) effects on perinatal outcomes by literature review and meta-analysis. Interesting results include that SMM is associated with increased neonatal mortality! I have no substantive comments to make. I only wish that there was more information on cause - like placental pathology or autopsy results, but that is another paper!
Author Response

(The authors gave the same response as above.)

Reviewer 3 Report
This paper presented a systematic review and meta-analysis of the literature on perinatal outcomes as impacted by maternal morbidity in high-income countries. This topic is of great importance, as many high-income countries are faced with growing rates of maternal morbidity and consequences, despite seeing a decrease in maternal and infant mortality. The study is well done overall. I have a few minor suggestions to help further contextualize the importance of the study.
Introduction Lines 74-83: Suggest emphasizing the costs of SMM in HICs to further justify why HICs were the focus of this study. Much attention in the MCH realm understandably goes to LMICs because of immediate need; however, the problem of SMM is still serious, even if it is not typically resulting in maternal mortality. Providing some information about the consequences of SMM to the healthcare system, economy, quality of life, etc. would be helpful to contextualize this issue.
Discussion Lines 276-287: Similar comment to the introduction; suggest providing more evidence about why this should be done, beyond the impact on individual's health. How does this affect the health care system?
Author Response

(The authors gave the same response as above.)
